# Association of Prolonged Fluoroscopy Time with Procedural Success of Percutaneous Coronary Intervention for Stable Coronary Artery Disease with and without Chronic Total Occlusion

**DOI:** 10.3390/jcm10071486

**Published:** 2021-04-03

**Authors:** Peter Tajti, Mohamed Ayoub, Thomas Nuehrenberg, Miroslaw Ferenc, Michael Behnes, Heinz Joachim Buettner, Franz-Josef Neumann, Kambis Mashayekhi

**Affiliations:** 1Department of Interventional Cardiology, Cardiology and Angiology II, University Heart Center Freiburg, 79189 Bad Krozingen, Germany; ptajti@gmail.com (P.T.); ayoub@hotmail.de (M.A.); Thomas.Nuehrenberg@universitaets-herzzentrum.de (T.N.); Miroslaw.Ferenc@universitaets-herzzentrum.de (M.F.); buettner@hahndorf.org (H.J.B.); franz-josef.neumann@universitaets-herzzentrum.de (F.-J.N.); 2Gottsegen György Hungarian Institute of Cardiology, 1096 Budapest, Hungary; 3First Department of Medicine, University Medical Center Mannheim, 68167 Mannheim, Germany; michael.behnes@umm.de

**Keywords:** complex coronary artery disease, chronic total occlusion, outcomes, percutaneous coronary intervention

## Abstract

Background: In percutaneous coronary interventions (PCI), the impact of prolonged fluoroscopy time (FT) on procedural outcomes is poorly studied. Methods and Results: We analyzed the outcomes of 12,538 consecutive elective PCIs. The primary endpoint was procedure failure (PF), the composite of technical failure, and adverse in-hospital events including all-cause death, myocardial infarction, stroke, and target vessel revascularization (MACCE), as well as pericardial tamponade. We stratified the procedures as PCI for chronic total occlusion (CTO, *n* = 2720) and PCI for non-CTO (*n* = 9818). Logistic regression demonstrated a significant association between fluoroscopy time and procedural failure with a significant interaction with PCI type (both *p* < 0.001). The odds ratios (OR) of procedural failure for a 10-min increment in FT were 1.15 (confidence interval (CI) 95% 1.12–1.18, *p* < 0.001) in non-CTO PCI and 1.05 (CI 95% 1.03–1.06, *p* < 0.001) in CTO PCI. The optimal cut-point for prediction of PF was 21.1 min in non-CTO PCI (procedural success in 98.4% versus 95.3%, adjusted OR for PF 2.79 (CI 95% 1.93–4.04), *p* < 0.001) and 41 min in CTO PCI (procedural success in 92.3% versus 83.8%, adjusted OR for PF 2.18 (CI 95% 1.64–2.94), *p* < 0.001). In CTO PCI, the increase in PF with FT was largely driven by technical failure (adjusted OR 2.25 (CI 95% 1.65–3.10), *p* < 0.001), whereas in non-CTO PCI, it was driven by major complications (adjusted OR 2.94 (CI 95% 1.93–4.53), *p* < 0.001). Conclusions: Prolonged FT is strongly associated with procedural failure in both non-CTO and CTO PCI. In CTO PCI, this relation is shifted towards longer FT. The mechanisms of procedural failure differ between CTO and non-CTO PCI.

## 1. Introduction

A wide range of complex coronary anatomy, including chronic total occlusions, bifurcations, and extreme calcification performed even in high-risk patients with severe comorbidities, are safely and efficiently manageable percutaneously [1]. The impact of complex PCI prolongation on procedural outcomes has been sparsely studied [2]. We sought to investigate the effect of fluoroscopy time on in-hospital major adverse cardiac and cerebrovascular events in patients undergoing elective PCI, with a goal of defining a cut-off time value that may advise operators considering procedure termination both targeting chronic occlusive and nonocclusive complex coronary lesions.

## 2. Materials and Methods

### 2.1. Study Population and Patient Selection

We analyzed the clinical, angiographic, and procedural characteristics of 16,317 consecutive PCIs performed in a referral PCI center at the Division of Cardiology and Angiology II, University Heart Center Freiburg—Bad Krozingen. Patients undergoing PCI for acute coronary syndrome were excluded (*n* = 3720), along with additional stable procedures in which fluoroscopy time was not recorded (*n* = 59). The final patient cohort included 12,538 PCI procedures performed with an overall 16,856 target vessels (Figure 1). The study was approved by the ethics committee of the Albert-Ludwigs-Universität Freiburg, Germany, (ID: EK 21-1100) and is in accordance with the ethical guidelines of the 1975 Declaration of Helsinki, as revised in 1983.

### 2.2. Endpoints and Definitions

The primary endpoint was procedural success as a composite of technical success without an in-hospital major adverse cardiac and cerebrovascular event (MACCE) and pericardial tamponade requiring either pericardiocentesis or surgery. Technical success was defined as successful revascularization of occlusive and nonocclusive coronary lesions with achievement of <30% residual diameter stenosis within the treated segment and restoration or maintenance of TIMI grade 3 antegrade flow. In-hospital MACCE included any of the following adverse events prior to hospital discharge: mortality, myocardial infarction, recurrent symptoms requiring urgent target vessel revascularization (TVR) and target lesion revascularization (TLR) with PCI or surgery, and stroke. Myocardial infarction (MI) was defined by the fourth Universal Definition (type 4a) described by Thygesen [3]. Sensitivity analysis for in-hospital MACCE included myocardial infarction based upon the Society of Cardiovascular Interventions and Angiography (SCAI) MI definition [4]. The *secondary endpoints* were components of the primary endpoint, coronary perforation managed conservatively, and Bleeding Academic Research Consortium (BARC) class 3 to 5 during the 30-day follow-up. Data collection on the 30-day short-term follow-up of patients who underwent PCI was obtained during office visits, via telephone contacts with the patient or family members, and careful assessment of medical records, as necessary.

All PCI procedures were performed in patients with stable angina. Coronary lesions were classified according to American College of Cardiology/American Heart Association Lesion Classification (ACC/AHA) to type A, B1, B2, and C lesions described by Ryan et al. [5]. Chronic total occlusion (CTO) was defined as a coronary lesion with Thrombolysis In Myocardial Infarction (TIMI) grade 0 flow of at least a 3-month duration as described by the 2019 Consensus Document of the EuroCTO Club [6]. Calcification was assessed by angiography as mild (spots), moderate (involving ≤ 50% of the reference lesion diameter), and severe (involving > 50% of the reference lesion diameter). Moderate proximal vessel tortuosity was defined as the presence of at least 2 bends > 70° or 1 bend > 90° and severe tortuosity as 2 bends > 90° or 1 bend > 120° in the target vessel. A CTO procedure was defined as “retrograde” if an attempt was made to cross the lesion through a collateral vessel or bypass graft supplying the target vessel distal to the lesion; if not, the procedure was classified as “antegrade”. Heparin was administered during the procedure maintaining and optimized activated clotting time of 250–300 s in non-CTO PCI and 300–350 for CTO PCI. All PCIs were performed by highly skilled operators, and PCI techniques or other treatments were left at the operator’s discretion.

### 2.3. Statistical Analysis

Categorical variables were expressed as percentages and were compared using Pearson’s chi-square test or Fisher’s exact test. Continuous variables were presented as mean ± standard deviation or median (interquartile range, IQR) unless otherwise specified and were compared using the t-test and one-way analysis of variance (ANOVA) for normally distributed variables; the Wilcoxon rank-sum test and the Kruskal-Wallis test were applied for nonparametric continuous variables, as appropriate.

The primary analysis tested the association between the primary endpoint and fluoroscopy time and its interaction with the type of PCI (CTO versus non-CTO) by logistic regression. Results are reported as odds ratios with 95% confidence intervals. Depending on the outcome of the primary analysis, we planned secondary analyses stratified by CTO versus non-CTO PCI. Receiver operating characteristic (ROC) curve analysis with application of the Youden-index was used to identify a cut-off of fluoroscopy time both in CTO and non-CTO PCI to predict procedural success (Appendix A). These cut-offs were used to examine the association between fluoroscopy time and primary and secondary endpoints both in non-CTO and CTO PCI subsets. Crude and adjusted odd ratios were calculated after selection of the confounding variables based on an univariable association with the given endpoints at *p* < 0.05. We also performed analyses of the primary endpoint, as well as of technical success and MACCE, according to quartiles of fluoroscopy time in the two strata defined by type of PCI. MACCE-free survivals during the 30 days of follow-up were calculated using the Kaplan–Meier method and compared between groups using the log-rank test. Adjusted Cox proportional hazard regression model was used to identify predictors of short-term MACCE and to calculate hazard ratios (HR). Confounding variables for the Cox proportional regression model were selected based on univariable association with MACCE at *p* < 0.05. As a sensitivity analysis, we repeated the primary analysis applying the SCAI definition of myocardial infarction (dependent variable). All statistical analyses were performed with JMP 13.0 (SAS Institute, Cary, NC, USA). A two-sided *p* value of 0.05 was considered statistically significant.

## 3. Results

### 3.1. Clinical, Angiographic and Procedural Characteristics of the Study Cohort

The baseline clinical features of the study population are summarized in Table 1, categorized according to procedural success (*n* = 12,000) and failure (*n* = 538). Incidence of diabetes mellitus, history of prior MI, and previous coronary artery bypass graft surgery (CABG) were less frequent in successful versus failed PCIs, whereas age, gender, left ventricular (LV) function, and New York Heart Association (NYHA) Class status remained similar in both groups.

The angiographic characteristics of the study lesions are presented in Table 2. Lesion complexity significantly differed between the compared groups; failed procedures were associated with longer lesion length and more severe calcification compared to successful interventions. CTO target vessel (50.8% vs. 15.6%, *p* < 0.001) and Type C AHA/ACC Complexity Class (68.6% vs. 39.0%, *p* < 0.001) lesions were more commonly observed in failed procedures.

Overall procedural and technical success were 95.7% and 97.3%, respectively. The baseline technical and procedural characteristics are presented in Table 3. Femoral access was selected in overall 38.3%, and use of rotational atherectomy device was 3.9%. CTO PCI was performed in 21.7% of all interventions. Antegrade wire escalation was the most commonly used approach (70.9%) in CTO interventions, and retrograde approach was applied in 29.1% of all CTO PCIs. Median contrast volume, dose area product, and procedure and fluoroscopy time were 190 (150–260) mL, 5364 (3146–9000) cGy*cm^2^, 42 (25–71) and 16.0 (9.0–28.0) minutes, respectively, and significantly differed between failed and successful procedures.

As shown in Table 4, fluoroscopy time was significantly associated with procedural failure. This relation showed a significant interaction with type of PCI, CTO versus non-CTO. Similar results were observed by applying the SCAI definition of MI as the component of in-hospital MACCE and procedural failure (both *p* < 0.001, Appendix A).

### 3.2. Secondary Analysis in Non-CTO PCI Subset

Overall, procedural and technical success were 97.7% and 99.3%, respectively. Baseline clinical and angiographic characteristics of successful and failed non-CTO PCI are described in Appendix A.

The odds ratio (OR) of procedural failure for a 10-min increment in FT was 1.15 (confidence interval (CI) 95% 1.12–1.18, *p* < 0.001). The optimal cut-point for prediction of PF was 21.1 min with an area under the curve (AUC) of 0.665 (*p* < 0.001) (Appendix A, Panel B). Procedural success was 98.4% below this cut-point and 95.3% above (adjusted OR 2.79 (CI 95% 1.93–4.04), *p* < 0.001). Prolongation of non-CTO procedures over the cut-point was independently associated with an increased chance for procedure failure both in univariate (OR 3.13, CI 95% 2.40–4.08, *p* < 0.001) and multivariate levels (OR 2.79, CI 95% 1.93–4.04, *p* < 0.001) (Figure 2).

Procedural and technical success significantly decreased and in-hospital MACCE significantly increased with quartiles of fluoroscopy time (Figure 3 Panel A). All secondary endpoints (Appendix A), except for stroke, vascular access complications, and contrast-induced nephropathy, showed a strong (*p* < 0.001 for all) association with quartiles of fluoroscopy time. There also was a weak but statistically significant association with vascular access complications (*p* = 0.013).

Prolongation of non-CTO PCI was also significantly associated with an increasing probability for in-hospital major complications (OR 2.96, CI 95% 1.93–4.56, *p* < 0.001), perforation (OR 4.34, CI 95% 2.53–7.60, *p* < 0.001), bleeding (OR 2.45, CI 95% 1.45–4.15, *p* < 0.001) development, and an increasing probability for technical failure (OR 2.10, CI 95% 1.19–3.71, *p* = 0.010) (Table 5). Prolonged non-CTO PCI was independently associated with a worse MACCE-free survival (log-rank test *p* < 0.001) compared to shorter PCIs (HR 2.08, CI 95% 1.41–3.05, *p* < 0.001) (Figure 4, Panel A).

### 3.3. Secondary Analysis of CTO PCI Subset

The overall procedural and technical success in CTO PCI were 88.5% and 89.7% and decreased with fluoroscopy time increments. The baseline clinical and angiographic parameters of successful and failed CTO interventions are summarized in Appendix A.

The odds ratios (OR) of procedural failure for a 10-min increment in FT were 1.05 (CI 95% 1.03–1.06, *p* < 0.001) in CTO PCI. The optimal cut-point for predicting procedural failure was >41.0 min of fluoroscopy time with an AUC of 0.613 (*p* < 0.001) (Appendix A, Panel C). Below the cut-point, procedural success was 92.3% as compared with 83.8% above the cut-point (*p* < 0.001). Multivariable logistics regression showed an independent association between CTO procedure prolongation and procedural success (OR 2.18, CI 95% 1.64–2.94, *p* < 0.001) (Figure 5). Likewise, prolonged CTO PCI above the cut-off was associated with an increased chance of technical failure and increased risks of perforation with and without tamponade and an increased risk of bleeding. However, prolongation of CTO-PCI above the cut-point was not independently associated with an increased risk for either the composite of procedure-related major complications (OR 1.53, CI 95% 0.89–2.65, *p* = 0.124) or 30-days MACCE post-PCI (HR 1.14, CI 95% 0.67–1.91, *p* = 0.626) (Table 6). Kaplan–Meier curves showed similar MACCE-free survivals irrespective of fluoroscopy time for CTO PCI (Figure 4, Panel B) (log-rank test *p* = 0.781).

Procedural and technical success significantly decreased with quartiles of fluoroscopy time (Figure 3 Panel B). Incidence of MACCE was not significantly different between quartiles of fluoroscopy time (1.86%, 1.19%, 1.05%, and 1.32%, *p* = 0.581). Incidence of perforation, pericardial tamponade (*p* < 0.001 for both), and bleeding (*p* < 0.001) showed significant increases with fluoroscopy time in the CTO cohort, whereas incidences of mortality, MI, stroke, TVR, and TLR were not significantly associated with fluoroscopy time in the CTO PCI cohort (Appendix A). Moreover, 30-day MACCE post-CTO PCI did not vary significantly across quartiles of fluoroscopy time (*p* = 0.763).

## 4. Discussion

To the best of our knowledge, our study is the largest to evaluate the impact of prolonged fluoroscopy time on procedural outcomes in both CTO and non-CTO PCI. Our major findings are as follows: (1) Prolonged fluoroscopy time is strongly associated with procedural failure. (2) The relation between fluoroscopy time and procedural failure differs significantly between non-CTO and CTO PCI, with a shift towards longer fluoroscopy times in CTO–PCI. (3) In non-CTO PCI, the decrease in procedural success with increasing fluoroscopy time is largely driven by MACCE, whereas with CTO PCI, it is driven by technical failure.

The procedural success significantly decreased with increasing fluoroscopy time in both CTO and non-CTO PCI. However, the optimal cut-off time for prediction of procedural failure in CTO PCI was about twice that in CTO PCI. Even an uncomplicated CTO PCI is inherently more time consuming than a non-CTO PCI, whereas in non-CTO PCI, prolongation of the procedure may be caused by the number of lesions treated or intraprocedural problems.

The mechanism of procedural failure differed substantially between the two types of PCI. Although the chance of technical failure increased significantly with fluoroscopy time in both modalities, the increment was small with non-CTO PCI—0.4% between first and fourth quartile of fluoroscopy time—but substantially larger with CTO PCI—7.3% between first and fourth quartile of fluoroscopy time. Likewise, the statistically significant increment in the risk of perforation with tamponade with fluoroscopy time was larger with CTO PCI than with non-CTO PCI (1.47% versus 0.49% for the difference between first and fourth quartile of fluoroscopy time). On the other side, the increase in procedural failure with FT in non-CTO PCI was driven by a significant increase in MACCE (2.37% increase between first and fourth quartile) that was not found in CTO PCI. The difference in MACCE depending on fluoroscopy time in the non-CTO group prevailed during 30-day follow-up, whereas in the CTO group, 30-day survival without MACCE was similar irrespective of procedure prolongation. Thus, in CTO PCI, the risk of MACCE was low and largely independent of fluoroscopy time. Possible explanations for this include the following: (a) a presumably higher level of tolerance for transient ischemia in patients undergoing PCI with a well-developed collateral system, (b) the predominant restriction of the procedure to one lesion, (c) a lower clinical impact of technical failure or early reocclusion, and (d) CTO operators’ advanced skillset for hindering procedure-related complications. Additionally, bleeding complications significantly increase in both longer CTO and non-CTO PCI. This may be caused by the cumulated dosage of unfractionated heparin, which is critical to maintain an appropriate activated clotting time (ACT). Other potential explanations include a higher rate of femoral access, larger sheath size, and potential blood loss via the Y-connectors caused by more frequent device exchange in lengthened PCI.

In the dichotomized analyses, fluoroscopy time above the cut-off was a strongly associated with procedural failure, roughly indicating a 2-fold increase in risk with CTO PCI and 3-fold increase with non-CTO PCI. Nevertheless, the discrimination was poor with AUCs of 0.665 (*p* < 0.001) and 0.613 (*p* < 0.001), respectively.

In prior studies, target lesion complexity has been associated with increasing need for PCI resources, such as contrast volume, radiation, and fluoroscopy time [7]. However, the reverse association between PCI prolongation and procedural outcomes—preferably using fluoroscopy time as indicator—has been rarely studied. Nikolsky et al. examined the relation between fluoroscopy time and short-term PCI prognosis in 9650 consecutive PCIs, in which 75th percentile threshold was utilized (18.3 ± 12.2 min) (2) as prolonged fluoroscopy time. Procedure prolongation was associated with an increased risk of in-hospital mortality (3.3% vs. 0.3%, *p* < 0.0001) emergent surgery (2.1% vs. 0.3%, *p* = 0.0001), and contrast-induced nephropathy (6.7% vs. 4.5%, *p* = 0.03). In a report from the United States National Cardiovascular Data Registry, Fazel et al. also analyzed the fluoroscopy time factor during invasive coronary angiography and percutaneous coronary intervention; however, they sought to evaluate the determinants of fluoroscopy time elongation but not its effect on procedural success and incidence of major complications related to PCI [8].

In previous studies, thresholds for defining procedure prolongation were simply selected as fluoroscopy time exceeding >75th percentile in the entire cohort (2,7). However, various procedures, such as CTO and non-CTO PCI, substantially differ in terms of techniques and clinical consequences. We, therefore, addressed the interaction of type of PCI (CTO versus non-CTO) with the impact of fluoroscopy time on procedural success in our multivariable logistic regression model. Based on a highly significant interaction as revealed by this model, we were able to calculate different cut-off values of procedure prolongation for CTO and non-CTO PCI. In a sensitivity analysis, we also confirmed that our findings were robust with respect to the definition of myocardial infarction. A similar association between fluoroscopy time and procedural success was present when implementing the SCAI definition instead of the fourth Universal Definition [9].

Our study has limitations. First, it has a retrospective, observational design without core laboratory assessment of the study angiograms or independent clinical event adjudication. Second, procedural complications, such as perforation, are self-reported; however, the occurrence of MACCE underwent a quality check performed by the dedicated independent committee of our institution. Third, study procedures were performed in a dedicated, high-volume CTO referral center. Fluoroscopy time thresholds and their effect on in-hospital outcomes may differ numerically in other cohorts. Fourth, our study was not designed to elucidate the underlying mechanism of the observed association between procedural failure and fluoroscopy time. To a large extent, prolonged fluoroscopy times are a reflection of the number and complexity of lesions treated, which both impact the risk of complications and technical failure. Therefore, this association is an epiphenomenon rather than responsible for success or failure of a procedure. On the other hand, excessive procedure duration may be detrimental in itself due to exhaustion of patients, toxic effects of contrast media, and operator fatigue.

## 5. Conclusions

In patients undergoing PCI for stable CAD, prolonged fluoroscopy time is associated with increased probability of procedural failure. Although this paradigm applies to both CTO and non-CTO procedures, CTO procedures are less sensitive to fluoroscopy time with regards to the risk of periprocedural MACCE, which is largely unaffected. In non-CTO PCI, however, the risk of periprocedural MACCE increases substantially with fluoroscopy time despite chances of technical success that are largely preserved. Thus, consideration of procedure termination is critical to reducing the risk of MACCE in non-CTO PCI. Further studies are required to identify a threshold for fluoroscopy time reliably, as often a procedure needs to be prolonged until completion with adequate flow to prevent potential catastrophic events. Nevertheless, when there is an option to safely abandon or stage a procedure, our findings can help operators in decision making.

## Figures and Tables

**Figure 1 jcm-10-01486-f001:**
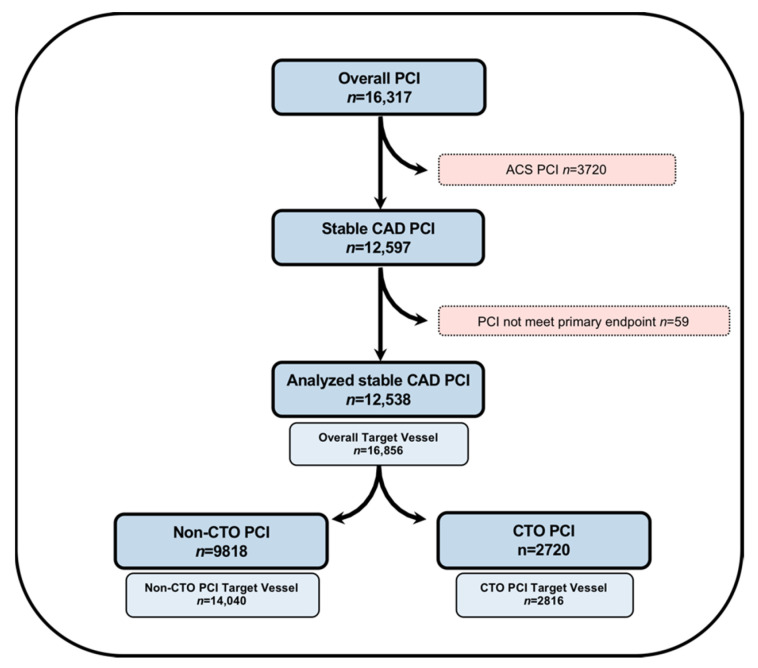
Flowchart of patient cohort selection in the current study. ACS, acute coronary syndrome; CAD, coronary artery disease; CTO, chronic total occlusion; and PCI, percutaneous coronary intervention.

**Figure 2 jcm-10-01486-f002:**
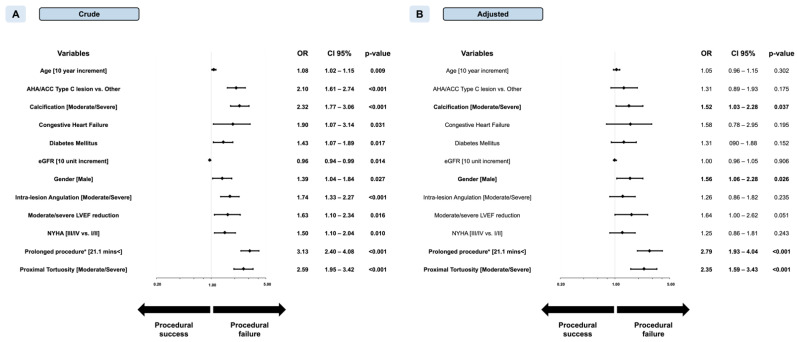
Forest plot of crude (Panel **A**) and adjusted (Panel **B**) odds ratios (OR) for prediction of procedural success in nonocclusive percutaneous coronary intervention (non-CTO PCI). ACC, American College of Cardiology; AHA, American Heart Association; CABG, coronary artery bypass graft; eGFR, estimated glomerular filtration rate; LVEF, left ventricular ejection fraction; MI, myocardial infarction; NYHA, New York Heart Association. * Indicates fluoroscopy time (min). Left ventricular function groups are indicated as follows: normal (52–100%) moderately reduced (41–51%), reduced (30–40%), and low (0–29%) in males; and normal (54–100%), moderately reduced (41–53%), reduced (30–40%), and severely reduced (0–29%) in females.

**Figure 3 jcm-10-01486-f003:**
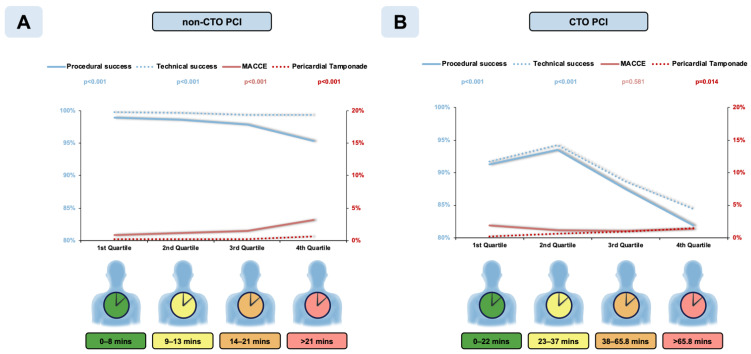
In-hospital outcomes in patients undergoing various length of percutaneous coronary interventions (PCI) targeting nonocclusive (Panel **A**) and chronic occlusive (Panel **B**) coronary lesions. CTO, chronic total occlusion, and MACCE, major adverse cardiac and cerebrovascular event.

**Figure 4 jcm-10-01486-f004:**
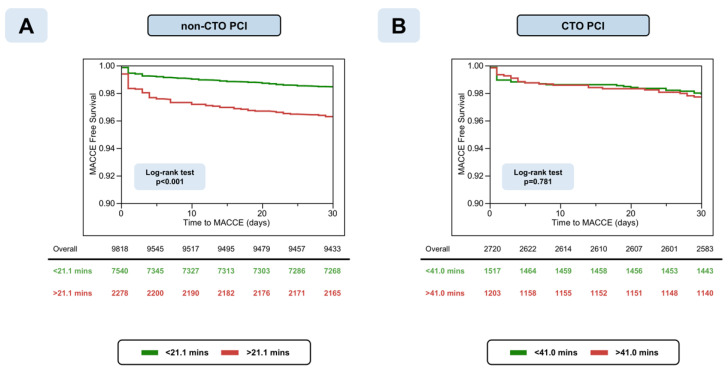
Kaplan–Meier curves of 30-days major adverse cardiac and cerebrovascular events (MACCE) in patients undergoing prolonged non-occlusive (non-CTO, Panel **A**) and chronic occlusive (CTO, Panel **B**) percutaneous coronary interventions (PCI).

**Figure 5 jcm-10-01486-f005:**
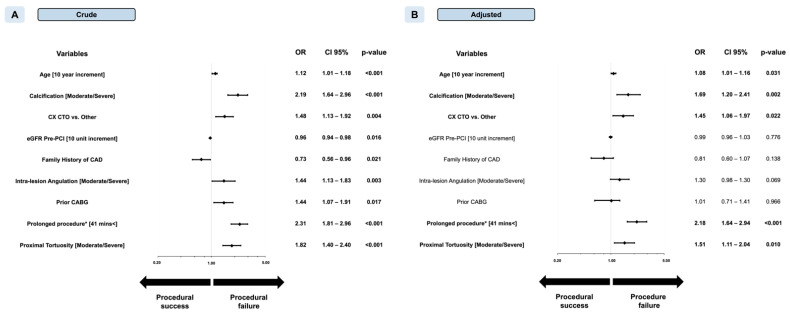
Forest plot of crude (Panel **A**) and adjusted (Panel **B**) odds ratios (OR) for prediction of procedural success in chronic total occlusion percutaneous coronary intervention (CTO PCI). CAD, coronary artery disease; CABG, coronary artery bypass graft; CTO, chronic total occlusion; CX, circumflex artery. * Indicates fluoroscopy time (min).

**Table 1 jcm-10-01486-t001:** Clinical characteristics of patient undergoing percutaneous coronary intervention (PCI) for stable coronary artery disease (CAD) categorized according to procedural outcomes.

Variable	Overall (*n* = 12,538)	Procedural Success (*n* = 12,000)	Procedural Failure (*n* = 538)	*p* Value
Age (years) *	69.3 ± 10.4	69.3 ± 10.4	69.9 ± 10.2	0.402
Men	77.2%	77.2%	77.5%	0.871
BMI (kg/m^2^) *	28.0 ± 4.6	28.0 ± 4.6	27.8 ± 4.7	0.342
Diabetes Mellitus	29.7%	29.4%	35.4%	0.005
Hypercholesterinemia	88.5%	88.4%	90.8%	0.103
Hypertension	86.8%	86.7%	89.0%	0.125
Smoking (current)	14.6%	14.5%	15.8%	0.413
Left Ventricular Ejection Fraction ^‡^				0.069
● Normal	71.9%	72.1%	66.6%	
● Moderately reduced	16.0%	15.9%	18.9%	
● Reduced	7.5%	7.5%	8.7%	
● Low	4.6%	4.5%	5.8%	
Family History of CAD	41.3%	41.5%	37.6%	0.099
Heart Failure	4.1%	5.6%	4.1%	0.099
NYHA Classification				0.523
● I	17.2%	17.2%	17.4%	
● II	47.7%	47.8%	44.4%	
● III	32.2%	32.1%	35.3%	
● IV	2.9%	2.9%	2.9%	
Prior MI	29.5%	29.3%	34.5%	0.015
Prior CABG	14.4%	14.0%	22.1%	<0.001
eGFR (mL/min/1.73^2^) *	71.3 ± 19.5	71.9 ± 19.4	69.4 ± 20.9	0.038
Baseline Creatinine (mg/dL) ^†^	1.00 (0.88, 1.20)	1.00 (0.88, 1.20)	1.00 (0.88, 1.22)	0.096

* mean ± standard deviation; ^†^ median (interquartile range). ^‡^ Left ventricular function groups are indicated as follows: normal (52–100%) moderately reduced (41–51%), reduced (30–40%), and low (0–29%) in males; and normal (54–100%), moderately reduced (41–53%), reduced (30–40%), and low (0–29%) in females. BMI, body mass index; CABG, coronary artery bypass graft; CAD, coronary artery disease; eGFR, estimated glomerular filtration rate; MI, myocardial infarction; NYHA, New York Heart Association.

**Table 2 jcm-10-01486-t002:** Angiographic characteristics of patients undergoing percutaneous coronary intervention (PCI) for stable coronary artery disease (CAD) categorized according to procedural outcomes.

Variable	Overall (*n* = 16,856)	Procedural Success (*n* = 16,208)	Procedural Failure (*n* = 648)	*p* Value
Target vessel				0.015
● RCA	27.8%	27.6%	32.4%	
● LAD	35.8%	36.1%	29.5%	
● LM	6.5%	6.6%	5.3%	
● CX	28.0%	27.8%	31.6%	
● Arterial Graft	0.2%	0.2%	0.0%	
● Venous Graft	1.8%	1.8%	1.2%	
Lesion Length				<0.001
● <10 mm	15.1%	15.3%	10.8%	
● 10–20 mm	43.8%	44.3%	31.7%	
● >20 mm	41.1%	40.4%	57.5%	
Calcification				<0.001
● None	7.2%	7.3%	3.8%	
● Mild (spots)	46.0%	46.8%	26.2%	
● Moderate	27.2%	27.3%	25.2%	
● Severe	19.6%	18.6%	44.8%	
Proximal Vessel Tortuosity	17.5%	17.0%	28.8%	<0.001
Intralesion Angulation				< 0.001
● None	11.3%	11.3%	10.7%	
● Discrete (<45)	47.2%	47.7%	34.5%	
● Moderate (45–90)	37.0%	36.7%	44.5%	
● Severe (>90)	4.5%	4.3%	10.3%	
CTO Target Vessel	17.0%	15.6%	50.8%	<0.001
Multiple Target Vessels	46.4%	46.5%	44.3%	0.265
In-stent Restenosis	12.7%	12.8%	10.6%	0.117
AHA/ACC Lesion Class				<0.001
● Type A	4.4%	4.5%	0.9%	
● Type B1	22.4%	22.9%	9.0%	
● Type B2	33.1%	33.6%	21.4%	
● Type C	40.1%	39.0%	68.6%	

ACC, American College of Cardiology; AHA, American Heart Association; CTO, chronic total occlusion; CX, circumflex artery, LAD, left anterior descending artery; LM, left main artery; RCA, right coronary artery.

**Table 3 jcm-10-01486-t003:** Technical and procedural characteristics of patients undergoing percutaneous coronary intervention (PCI) for stable coronary artery disease (CAD) categorized according to procedural outcomes.

Variable	Overall (*n* = 12,538)	Procedural Success (*n* = 12,000)	Procedural Failure (*n* = 538)	*p* Value
Femoral Access Used	38.3%	37.5%	57.4%	<0.001
Guide Catheter Size *	6.2 ± 0.5	6.1 ± 0.5	6.2 ± 0.5	<0.001
Rotational atherectomy	3.9%	3.9%	4.5%	0.519
CTO PCI	21.7%	20.1%	58.0%	<0.001
CTO Crossing Strategies Used ^‡^				0.082
Antegrade only	70.9%	71.4%	66.7%	
Retrograde	29.1%	28.6%	33.3%	
Number of DES Used ^†^	1 (1, 2)	1 (1, 2)	0 (0, 1)	<0.001
Overall Stent Length ^†^	24 (16, 38)	24 (16, 38)	28 (18, 52)	0.078
Technical Success	97.3%	100.0%	35.9%	<0.001
Length of Hospital Stay (days) *	2 (2, 2)	2 (2, 2)	2 (2, 4)	<0.001
Procedural Time (min) ^†^	42 (25, 71)	41 (24, 68)	82 (46, 134)	<0.001
Fluoroscopy Time (min) ^†^	16.0 (9.0, 28.0)	15.0 (9.0, 26.4)	34.0 (17.0, 65.0)	<0.001
Contrast Volume (mL) ^†^	190 (150, 260)	180 (150, 250)	250 (170, 380)	<0.001
Dose Area Product (cGy*cm2) ^†^	5364 (3146, 9000)	5299 (3124, 8790)	8554 (4333, 15,422)	<0.001

* mean ± standard deviation; ^†^ median (interquartile range). ^‡^ Only applicable for CTO PCI. CTO, chronic total occlusion; DES, drug eluting stent; FFR, fractional flow reserve; iFR, instant wave-free ratio; PCI, percutaneous coronary intervention.

**Table 4 jcm-10-01486-t004:** Primary logistic regression analysis between chronic occlusive (CTO) percutaneous coronary intervention (PCI), fluoroscopy time (both as independent variables), and procedural failure (as dependent variable) by applying the fourth Universal Definition (UD) of myocardial infarction (MI).

Variable	Procedural Failure (4th UD MI)		
	OR	CI 95%	*p* Value
CTO PCI (vs. non-CTO PCI)	3.67	2.99–4.50	<0.001
Fluoroscopy time (mins)	1.02	1.02–1.02	<0.001
CTO PCI and Fluoroscopy time (mins)	-	-	<0.001

**Table 5 jcm-10-01486-t005:** Multivariate analysis between secondary endpoints (technical success, major complications, perforation, bleeding, 30-days MACE) and prolonged nonocclusive (non-CTO) percutaneous coronary intervention (PCI) according to the predefined cut-off minute value (<21.1 min).

Variable	Nonadjusted			Adjusted		
	OR	CI 95%	*p* Value	OR	CI 95%	*p* Value
Technical failure	3.34	2.05–5.44	<0.001	2.10	1.19–3.71	0.010
Major complications *	2.99	2.21–4.04	<0.001	2.96	1.93–4.56	<0.001
Perforation	4.94	2.98–8.31	<0.001	4.34	2.53–7.60	<0.001
Bleeding	2.77	1.90–4.03	<0.001	2.45	1.45–4.15	<0.001
30-days MACCE ^†§^	2.41	1.82–3.19	<0.001	2.08	1.41–3.05	<0.001

* Composite of in-hospital MACCE and pericardial tamponade requiring either pericardiocentesis or surgical evacuation. ^†^ Composite of all-cause death, myocardial infarction, stroke, target vessel-, and lesion revascularization. ^§^ Indicates hazard ratio (HR).

**Table 6 jcm-10-01486-t006:** Multivariate analysis between secondary endpoints (technical success, major complications, perforation, bleeding, 30-days MACE) and prolonged chronic occlusive (CTO) percutaneous coronary intervention (PCI) using the predefined cut-off minute value (<41.0 min).

Variable	Nonadjusted			Adjusted		
	OR	CI 95%	*p* Value	OR	CI 95%	*p* Value
Technical failure	2.16	1.70–2.79	<0.001	2.25	1.65–3.10	<0.001
Major complications *	1.51	0.90–2.59	0.120	1.53	0.89–2.65	0.124
Perforation	3.40	2.00–6.03	<0.001	3.07	1.76–5.58	<0.001
Bleeding	3.94	2.30–7.09	<0.001	4.05	2.25–7.61	<0.001
30-days MACCE ^†§^	1.10	0.65–1.84	0.719	1.14	0.67–1.91	0.626

* Composite of in-hospital MACCE and pericardial tamponade requiring either pericardiocentesis or surgical evacuation. † Composite of all-cause death, myocardial infarction, stroke, target vessel, and lesion revascularization. ^§^ Indicates hazard ratio (HR).

## Data Availability

Our study does not provide publicly available datasets.

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
