# Peer review of "Association of Prolonged Fluoroscopy Time with Procedural Success of Percutaneous Coronary Intervention for Stable Coronary Artery Disease with and without Chronic Total Occlusion"

_jcm, 2021, doi:10.3390/jcm10071486_

Round 1
Reviewer 1 Report
Manuscript ‚Impact of prolonged fluoroscopy time on procedural success of percutaneous coronary intervention for stable coronary artery disease with and without total chronic occlusion’.
I congratulate the authors on the interesting manuscript, I have only few questions and comments.
- I am not sure, whether ‘Impact of fluoroscopy time on procedural success …’ is the right title for the manuscript. It may be better to talk of the association of fluoroscopy time and procedural success. In the first chapter of the Discussion the authors themselves use the word predictor.
- This is because fluoroscopy time is probably rather an epiphenomenon in the whole process than itself responsible for success or failure.
- The different associations of CTO and non-CTO cases is interesting. Did experience of the investigators play a role? A short comment in Materials and Methods concerning this matter could clarify.
Author Response
Reviewer #1
I congratulate the authors on the interesting manuscript, I have only few questions and comments.
Response:We greatly appreciated the reviewer’s positive feedback.
Comment #1:I am not sure, whether ‘Impact of fluoroscopy time on procedural success …’ is the right title for the manuscript. It may be better to talk of the association of fluoroscopy time and procedural success. In the first chapter of the Discussion the authors themselves use the word predictor.
Response #1:We appreciate the reviewer’s suggestion on modifying the title of the manuscript. We changed the title to ‘Association of prolonged fluoroscopy time with procedural success of percutaneous coronary intervention for stable coronary artery disease with and without chronic total occlusion’. We also removed the phrase ‘predictor’ in the discussion section.
Comment #2:This is because fluoroscopy time is probably rather an epiphenomenon in the whole process than itself responsible for success or failure.
Response #2:We appreciate the insightful comment. We added a sentence in the limitation section describing that fluoroscopy time prolongation is rather the epiphenomenon, than the responsible factor of procedure failure or success.
Comment #3:The different associations of CTO and non-CTO cases is interesting. Did experience of the investigators play a role? A short comment in Materials and Methods concerning this matter could clarify.
Response #3:We greatly appreciate the reviewer’s thoughtful comment on the operators skillset and experience. All PCIs were performed by highly skilled operators, nevertheless, use of PCI techniques were left at the operators discretion. We added a sentence to the Materials and Methods section as suggested.

Reviewer 2 Report
The manuscript is a very detailed retrospective analysis performed on the large amount of data from an experienced interventional center.
The aim was formulated as impact of the lenght of fluoroscopy on the procedural outcomes. Authors conclude that fluoroscopy time predicts of the outcomes. In my opinion the fluo time is NOT the predictor. A predictor (as the name states) should be estimated before the procedure, it is prospective, while fluo time is retrospective. Thus it should not be used in this context.
Author Response
Reviewer #2
The manuscript is a very detailed retrospective analysis performed on the large amount of data from an experienced interventional center.
Response:We greatly appreciated the reviewer’s positive comments.
Comment #1:The aim was formulated as impact of the length of fluoroscopy on the procedural outcomes. Authors conclude that fluoroscopy time predicts the outcomes. In my opinion the fluoro time is NOT the predictor. A predictor (as the name states) should be estimated before the procedure, it is prospective, while fluoro time is retrospective. Thus it should not be used in this context.
Response #1: We thank the reviewer the insightful comment and suggestion. We changed the address to ‘Association of prolonged fluoroscopy time with procedural success of percutaneous coronary intervention for stable coronary artery disease with and without chronic total occlusion’, and also removed the phrase ‘predictor’ and exchanged it to ‘associated with’.
